# ML-Enhanced Live Video Streaming in Offline Mobile Ad Hoc Networks: An Applied Approach

Manuel Jesús-Azabal [1,*] , Vasco N. G. J. Soares [2,3,4,*] and Jaime Galán-Jiménez [1]

1 Departamento de Ingeniería Sistemas Informáticos y Telemáticos, Escuela Politécnica de Cáceres, Universidad de Extremadura, Avenida de Elvas, S/N, 06006 Badajoz, Extremadura, Spain; jaime@unex.es
2 Polytechnic Institute of Castelo Branco, Av. Pedro Álvares Cabral, n° 12, 6000-084 Castelo Branco, Portugal
3 Instituto de Telecomunicações, Rua Marquês d'Ávila e Bolama, 6201-001 Covilhã, Portugal
4 AMA—Agência para a Modernização Administrativa, Rua de Santa Marta, n° 55, 1150-294 Lisboa, Portugal
* Correspondence: manuel@unex.es (M.J.-A.); vasco.g.soares@ipcb.pt (V.N.G.J.S.)

**Abstract:** Live video streaming has become one of the main multimedia trends in networks in recent years. Providing Quality of Service (QoS) during live transmissions is challenging due to the stringent requirements for low latency and minimal interruptions. This scenario has led to a high dependence on cloud services, implying a widespread usage of Internet connections, which constrains contexts in which an Internet connection is not available. Thus, alternatives such as Mobile Ad Hoc Networks (MANETs) emerge as potential communication techniques. These networks operate autonomously with mobile devices serving as nodes, without the need for coordinating centralized components. However, these characteristics lead to challenges to live video streaming, such as dynamic node topologies or periods of disconnection. Considering these constraints, this paper investigates the application of Artificial Intelligence (AI)-based classification techniques to provide adaptive streaming in MANETs. For this, a software-driven architecture is proposed to route stream in offline MANETs, predicting the stability of individual links and compressing video frames accordingly. The proposal is implemented and assessed in a laboratory context, in which the model performance and QoS metrics are analyzed. As a result, the model is implemented in a decision forest algorithm, which provides 95.9% accuracy. Also, the obtained latency values become assumable for video streaming, manifesting a reliable response for routing and node movements.

**Keywords:** mobile ad hoc networks; live-video streaming; artificial intelligence; machine learning; bluetooth low energy; offline streaming





## 1. Introduction

Video streaming has been established as one of the primary uses of communication networks [1]. It is estimated that 65.93% of current Internet traffic is related to video streaming [2], underscoring the significance of these applications in today's digital landscape. In the realm of real-time video streaming, a multitude of applications has emerged for purposes such as surveillance, entertainment, autonomous driving, text and object recognition, and social communications [3]. These services meet strict requirements for Quality of Service (QoS), which refers to the overall performance of network service parameters, and Quality of Experience (QoE), which denotes the overall satisfaction of the user experience, necessitating minimal delay and negligible interruptions [4]. In this context, technologies like 4G or 5G, and architectures such as cloud computing have provided significant advancements [5].

The improvement in communication technologies and cloud processing techniques has enabled an effective integration of real-time video flows into applications [5]. Transmission protocols such as HTTP Live Streaming (HLS) [6] or Dynamic Adaptive Streaming over HTTP (DASH) are applied in cloud architectures to provide large-scale broadcasting, while meeting the high QoS requirements [7]. However, there are contexts in which these

architectures are not applicable due to restrictions in connectivity, lack of infrastructure availability, or exaggerated required investment. In these scenarios, alternatives such as Mobile Ad hoc Networks (MANETs) become suitable approaches.

MANETs are a network typology which integrates autonomous independent mobile devices as nodes to perform data transmissions [8]. In these networks, communications are handled by the nodes using range-limited wireless communication interfaces, without requiring a centralized coordinator element. Nodes may change their position in the context, dynamically changing the network topology and creating additional connectivity opportunities [8]. As a result, MANETs become applicable in scenarios in which Internet infrastructure is not available, relying in the autonomy and mobility of nodes to perform data transmissions. Thus, MANETs may provide a suitable alternative for live video streaming in offline contexts; however, they face challenging limitations.

The inherent autonomy of the nodes in MANETs raises multiple challenges which may compromise QoS and QoE in the context of live video streaming [9]. The nodes mobility and the dynamic topology may lead to abrupt variations in the Received Signal Strength Indicator (RSSI) of nodes, which measures the power present in a received radio signal, affecting QoS by decreasing the bit rate, increasing latency, and, eventually, causing transient disconnections [9,10]. At the same time, nodes face power consumption and computing limitations, which may constrain their functions when processing incoming live video or acting as intermediate devices to route video streams [9]. Considering these challenges, the present work proposes a software-driven architecture to adapt live video streaming using Machine Learning (ML) in offline MANETs.

The proposal enables the transmission and routing of real-time video, adapting the image compression individually to the network state. To this end, ML classification techniques are applied to identify the stability of the link. Following this description, an implementation of the architecture has been developed and assessed in a laboratory context, training and applying a decision tree algorithm to achieve link quality prediction and adapting compression accordingly. As a result, QoS and QoE have been successfully satisfied.

The remainder of this paper is structured as follows: Section 2 describes related works and compares them with the proposed technique. Next, Section 3 details the functioning of the architecture and its components. Then, Section 4 assesses the implementation and analyzes the evaluation score of the model in terms of QoS and QoE. Finally, Section 5 draws draws conclusions and presents suggestions for further studies.

## 2. Related Works

The inherent nature of MANETs implies multiple challenges for delivering video, requiring measures to mitigate their impact on QoS. Some of their features often become obstacles to video streaming, such as the changing topology, node mobility, or limited bandwidth [9,11]. These factors lead to variations in RSSI, increased latency, decreased bit rates, and even route interruptions with the sender node. Consequently, live video streaming may experience delays and lack synchronization at destination nodes, while also encountering interruption periods.

In response to the challenge of streaming video in MANETs, the approaches presented in the literature may primarily focus on the routing process [12–15], while others provide adjustments to enhance performance. Thus, these works can essentially be classified into three groups [11]: multipath routing, QoS-aware routing, and prediction-based link routing.

Multipath routing is a technique based on maintaining multiple data flows towards a destination, with the aim of improving QoS [9]. To achieve this, multiple flows are employed to transmit a compressed version of the video stream and an enhanced version of the frames, respectively, through independent paths. Simultaneously, the use of several connections enables the aggregation of additional bandwidth while balancing the traffic load between nodes [11]. Furthermore, the use of multiple paths reduces the potential impact of link failures, thanks to redundancy. Proposals such as [16,17] represent some of the most significant alternatives in the field, applying resources like distance-vector

and memetic algorithms to enhance multimedia communication. Despite these advances, multipath streaming requires additional computational resources to prevent duplicated data at destinations [11].

On the other hand, QoS-aware techniques focus on adapting video streaming to the estimation of the available network resources [11]. Thus, the transmission rate is successfully adapted to the link state. Works such as [18,19] predict the required and available bandwidth to discover the optimal route toward the destination. To achieve this, a heuristic approach is applied, enabling the initial discovery of optimal paths. However, most of these QoS solutions focus on the optimization of low-level mechanisms, requiring specific underlying technologies for encoding and decoding. Therefore, most of them are not applicable for MANETs based on interfaces such as radio frequencies or Bluetooth.

Link-prediction routing is based on the estimation of a path between the sender and the destination to forward information [20]. The most recurrent techniques applied to predict stable links include periodic patterns, decision trees, complex network predictions, deep learning, and reinforcement learning techniques [21].

In the case of applying pattern-based predictions, models can be trained to predict links using historical contacts [20]. Thus, by identifying patterns in past encounters, link stability can be forecasted. Proposals like [22] explore the potential of this technique in networks, incorporating variables such as encounter time, recurrence, and distribution. However, the dynamic topology of MANETs often does not mirror social behaviors but is characterized by random and opportunistic encounters instead. As a result, this technique may exhibit limitations in its application to these contexts.

Decision trees are a potential classification mechanism that can be used to predict the quality of links. Relevant works such as [23] utilize this technique to determine paths towards the destination. For this purpose, attributes such as the speed of the node, the link expiration time, the trip time, and the node lifetime are applied to classify encounters. However, the considered metrics may not be accurate for real-time video transmission, since essential parameters such as latency are not considered.

In the case of deep learning, neural networks are applied to classify links and determine eventual stable connections. Works such as [24] utilize this technique to establish a graded relationship between nodes, facilitating the discovery of potential paths in networks and eventual recovery in case of link failures. However, its potential application for live video streaming remains limited. Conversely, contributions such as [25] showcase the utility of these technologies for audio streaming.

Reinforcement learning is conceived as a ML technique based on evaluating the actions of the model and rewarding it for making good decisions [26]. Proposals such as the one described in [26] rely on reinforcement learning to provide routing based on energy consumption in MANETs, enabling its integration for data transmissions. To this end, variables such as node energy levels, neighbor count, and traffic load are defined as the state space, while routing becomes the available action. Then, the reward is defined as a function which quantifies energy efficiency. However, in spite of these advancements, Ref. [26] is not adapted to video streaming.

After summarizing some of the most relevant related works, it is possible to highlight the most significant points. Video streaming in MANETs has mainly motivated works related to the routing process. Among the different approaches, ML-based ones have become the most relevant, offering adaptable functioning that enables effective evaluation of link stability. As a result, the application of these techniques may present a potential opportunity to enhance the performance of the proposed architecture.

## 3. ML-Driven Adaptive Streaming Quality in MANETs

The work presented in this paper proposes a software-based application to enable adaptive live video transmission in MANETs. To this end, a ML classification model is applied to evaluate the stability of links, enabling an individual adaptation of the image quality for each connection. Thus, the model makes use of communication metrics to

classify live connections, enabling the architecture to adapt the compression rate of the transmitting frames. As a result, frames are largely compressed when the connection becomes unstable. In this section, the working scheme of the proposal, the architecture, the routing process, and the ML model are detailed.

### 3.1. Working Scheme

The proposed architecture follows two main stages which are regularly executed: scanning and analyzing the network context, and adapting video streaming to contacted nodes. The combination of these tasks enables an accurate analysis of the nodes available in the context, while dynamics communications are carried out.

Figure 1 depicts an initial scenario for a MANET, including the initial setup of the architecture. In this case, four nodes are located in the context: Sender, Node A, B and C. The Sender node is in charge of broadcasting live video streams, while the remaining nodes are spectators. In order to scan and evaluate the network, the Sender transmits multiple pings to the surrounding nodes (step 1) and waits for their response (step 2). This way, it is possible to register multiple QoS variables such as the above-mentioned RSSI; throughput, which is the rate of successful message delivery over a channel; Bit Error Rate (BER), which is the percentage of bits that have errors relative to the total received; number of connection attempts; latency; and distance. These metrics are used by the model to evaluate the link stability (Table 1), predicting if the current connection is stable or unstable. As a result, frames in the video streaming can be compressed accordingly.

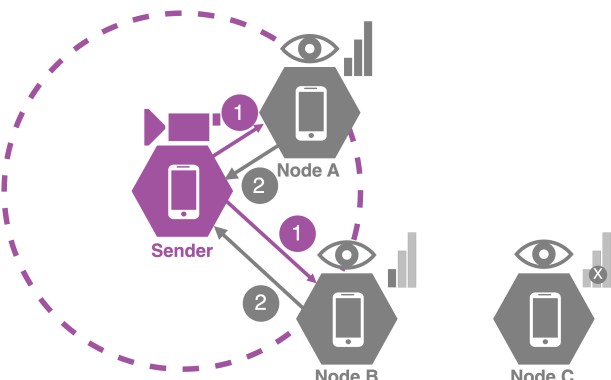

**Figure 1.** Initial setup in the proposed architecture.

**Table 1.** Attributes considered in the ML model to predict link quality.

| Parameter | Value |
| --- | --- |
| RSSI | Strength of the wireless signal between two nodes. Considers the influence of distance and physical objects of the context. |
| Throughput | Volume of data transmitted successfully. Actively depends on the stability of the link. |
| BER | Proportion of error bits regarding the total amount of bits. A high BER indicates eventual problems in the signal quality. |
| Number of connection attempts | Frequency with which the connection goes down and needs to reconnect. An increasing number of connection attempts may represent weak stability in the link. |
| Latency | Time elapsed between the data is transmitted and it is eventually received. High latency values may affect QoS in the stream. |
| Distance | Distance estimation between sender and receiver. High values of distance may affect link stability. |

Figure 2 illustrates the progression of the first scenario and the subsequent steps. After the nodes have been identified by the sender and their stability has been classified, adaptive video streaming is transmitted to the connected nodes (step 3). Specifically, Node

A receives frames with low compression due to its stable connection. Conversely, due to its unstable connection, Node B receives highly compressed frames. Initially, Node C is not recognized by the sender. Nonetheless, other nodes conduct pings to surrounding devices and evaluate the responses (step 4). In this scenario, Node B pings both A and C but only receives a response from C, as nodes do not accept more than one concurrent incoming communication if their current link is classified as stable. Consequently, Node C is recognized as a new node and receives video streams, which are adapted by the intermediate node (step 5). This process facilitates routing in MANETs, with dynamic adaptation of compression.

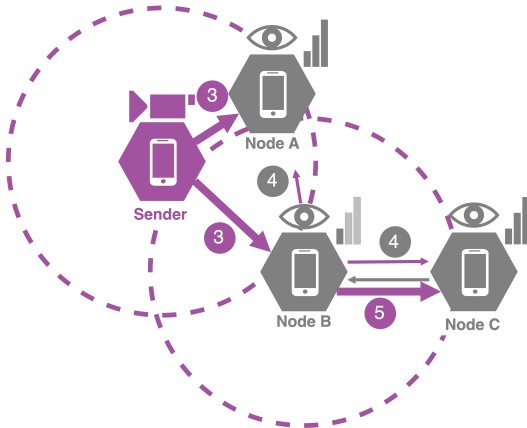

**Figure 2.** Adaptative video streaming after context analysis.

These advances have been made possible through the coordination of multiple transmission processes. Therefore, the proposed architecture comprises multiple elements dedicated to communication management, link prediction, and frame forwarding. Taking this into consideration, the following subsection will analyze the internal elements that constitute the proposal.

*3.2. Architecture*

The proposed application relies on the integration of various software-based components, facilitating dynamic and individual adaptability of streaming. As depicted in Figure 3, the functionality of the involved elements varies based on the node's role. A device assumes the sender role when it is responsible for capturing live video, whereas it becomes a destination when receiving and displaying the live video streaming. Additionally, these nodes play a crucial role in routing the streaming to destination devices that are not directly reached by the sender.

The sender node performs three main tasks: scanning and analyzing the network context; capturing live video; adapting frame compression and delivering frame.

1.  Scanning and Analyzing the Network Context (Communication Module): The sender node monitors surrounding devices that announce their presence, identifying them as nodes within the MANET. Subsequently, the sender pings these devices to collect a set of performance metrics, which are utilized to classify the connection's stability. A pre-trained ML-based classification model processes these parameters to determine if the link is stable. The insights gained from this scanning process are then used to adjust frame compression. Therefore, scanning and analysis are conducted periodically to keep the link states updated.
2.  Capturing Live Video: Video streaming is initiated by the sender node, which captures a series of frames comprising the video flow. A lightweight buffer is employed to retain frames during the transmission process.
3.  Adapting Frame Compression and Delivering Frames: When the sender captures a frame, its compression can be adapted according to the predicted stability of the communication. This adjustment considers the latest prediction stored for each connection.

In this context, video and image compression algorithms enable the specification of parameters that vary the compression rate of the stream. However, it is important to consider the potential impact of these techniques on the constrained resources of MANET nodes, as well as their compatibility with the devices. Considering this, appropriate compression techniques should be available for most devices and require low computation [27]. Thus, algorithms such as H.264/Advanced Video Codec (AVC) become the most suitable option, due to their efficiency and robust performance [27]. This technique offers a balance between quality and file size, without implying additional overhead and redundancy. Another approach is compressing frames as images, applying codifications such as JPEG [28]. This technique has achieved suitable results for video transmission in constrained communication interfaces and enables an agile adaptation of the quality [28]. As a result, this technique may be applied in the adaptive streaming transmission, utilizing the output of the prediction model to adjust frames accordingly.

Alongside these processes, destination nodes undertake a series of tasks aimed at receiving video streams and simultaneously acting as intermediate nodes for devices beyond the sender's direct reach. These tasks encompass three main activities: scanning and analyzing the context, screening the video, and re-transmitting the video.

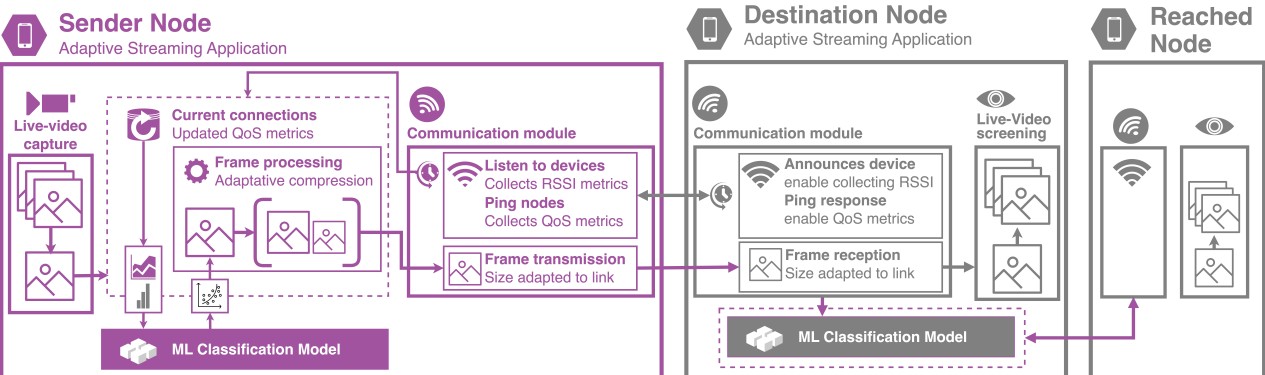

**Figure 3.** Architecture to dynamically adapt live video streaming in MANETs.

1.  Scanning and Analysis of the Network: Employing a strategy akin to that used by the sender, nodes conduct selective pings to assess the quality of their links. Consequently, the pool of connections is narrowed down to nodes that are not designated as senders. Furthermore, surrounding nodes only accept one concurrent incoming connection if their existing link is either predicted to be stable or originates directly from the sender. This policy is devised to prioritize traffic from the sender. Consequently, pings that are accepted are then utilized to gather metrics and forecast the quality of active links. To ensure this information remains current, this procedure is executed periodically.
2.  Screening Video: After receiving the frames from the sender, the receiver stores these images in a flash buffer. This setup facilitates video screening for the user, allowing them to view the stream while concurrently re-transmitting content to other nodes. Given that images are dynamically compressed at the source, the resolution of the received stream may vary, reflecting the quality of the link.
3.  Re-transmitting Video: Drawing on the data acquired during the scanning and analysis phase, the node may adjust the compression of frames based on the anticipated link quality. Nonetheless, with the intention of circumventing excessive compression of frames—which could diminish legibility after several hops—a minimum threshold is established.

These steps leverage the dynamic nature of MANETs to enable agile communications without the need for additional infrastructure. However, utilizing the individual resources of the nodes leads to a significant scalability limitation due to computational constraints

and energy consumption [8]. These factors present common challenges in MANETs and require extra measures to mitigate their impact on performance.

In response to the inherent challenges of MANETs, the proposed architecture addresses scalability by limiting potential connections in routing. This approach facilitates management in dense environments in which a large number of nodes are interconnected. Accordingly, receiver nodes are configured to accept new connections only when they are not currently connected or when the incoming link offers greater stability than the existing one. Consequently, this strategy helps prevent multiple nodes from routing streams to the same destinations, ensuring that priority is given to more stable links. Conversely, the Sender and the intermediate nodes are designed to route streaming to multiple devices, thereby incorporating an additional layer of scalability management.

Unlike the reception process, sender and intermediate nodes can transmit video to multiple destinations. In such cases, networks may experience high densities with a large number of nodes participating in communication. To prevent excessive consumption of computational resources and energy, scalability can be managed by establishing a threshold for the maximum number of connections. This limit ensures that each node can support an appropriate number of concurrent outgoing connections. However, this restriction is primarily enforced in ultra-dense contexts, as the limitations on reception are generally adequate to manage most scenarios.

Similar to the challenges of scalability, security emerges as a significant concern in MANETs. The essential connectivity between devices, coupled with the use of local wireless interfaces, demands the incorporation of mechanisms dedicated to ensuring security and averting potential attacks [29]. In the context of the proposed architecture, it is especially critical to address security in a manner that preserves the QoS and QoE. Thus, security measures must be designed to adapt to the network's evolving topology.

Given that the architecture enables streaming transmission within dynamic MANET environments, emphasis on security is essential for ensuring both the privacy and integrity of the streamed content. To this end, a hybrid approach that combines symmetric and asymmetric encryption can satisfactorily meet the stringent QoS requirements, while aiming to reduce its impact on the resource consumption in nodes [29,30]. Concretely, the symmetric encryption technique uses the same key for both encryption and decryption, becoming efficient in MANETs. However, it presents the challenge of keeping the key confidently. On the other hand, asymmetric encryption involves a pair of keys: a public key and a private key. Data encrypted with the public key can only be decrypted with the corresponding private key, and vice versa. This allows for secure communication in which the public key can be shared openly [29]. Considering this, a hybrid approach can be applied, aiming to apply asymmetric encryption to share a symmetric key between ends.

As Figure 4 shows, the proposed architecture utilizes the asymmetric logic to distribute the keys between the Sender and receivers, enabling them to maintain secure communications between them. Upon confirmation of a ping, the Sender distributes its public key to the receiver node, which answers with its public key (step 1). This technique enable the encryption of a symmetric key to encrypt video communication. In this case, the Sender encrypts the symmetric key with its private key, enabling the receiver to use the received public key to decrypt it. Once keys are shared, the Sender node encrypts the outgoing stream with the symmetric key, which has already been received by the node (step 2). As a result, the receiver node can decrypt the streaming with the symmetric key. Similarly, when a receiver node confirms a ping from another node for routing purposes, they begin a process of sharing their public keys (step 3), enabling the transmission of the symmetric key, originally generated by the Sender, to encrypt and decrypt the video stream. This method integrates asymmetric sharing versatility with symmetric encryption, aiming to minimize the potential impact on QoS, while avoiding expensive computational overload [30]. However, it is important to consider that this option is very sensitive to scalability. Thus, in the case of ultra-dense contexts, symmetric encryption becomes an optimal option when dealing with a large number of nodes.

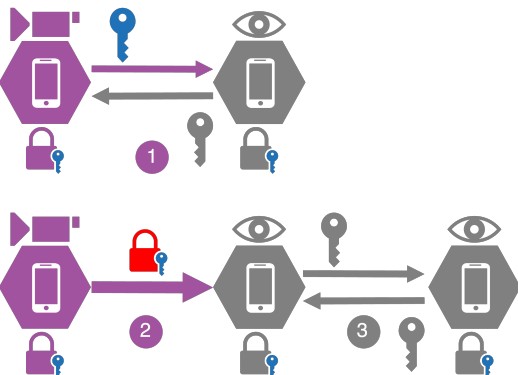

**Figure 4.** Hybrid encryption combining asymmetric technique to share symmetric keys.

Given the importance of the stages outlined, routing emerges as a crucial step for nodes within the architecture. Consequently, the following subsection will delve into the strategy employed and the principal decisions made regarding routing.

### 3.3. Routing Policy

Streaming in MANET networks presents a significant challenge due to frequent disconnections between nodes. The architecture introduces a strategy for establishing connections that focuses on prioritizing high-quality links and direct contact with the sender. Receiver nodes act as intermediate devices within the MANET, detecting nearby devices through pinging and assessing communication metrics. To maintain more stable connections, nodes that are already connected to the sender or another node refrain from accepting additional incoming transmissions. However, this approach may lead to nodes rejecting new links that could potentially offer greater stability than the existing ones. Therefore, it is crucial to establish a policy that ensures the optimization of current connections, guaranteeing that they are the most stable option available.

Figure 5 illustrates the routing policy implemented in the architecture, including the steps of stream retransmission. Thus, the context involves three nodes (Node A, B and C) and the sender. Initially, Node C is reachable only through Node B, which acts as a relay for video streaming transmission (step 1). Over time, Node C moves closer to Node A (step 2), resulting in Node A recognizing this link as stable (step 3). Concurrently, as Node B distances itself, its link with Node C is deemed unstable (step 4). Consequently, Node C has the option to switch to the newly established stable link with Node A, thereby disconnecting from Node B (step 3). This dynamic process underscores the architecture's ability to adaptively manage connections based on evolving link stability.

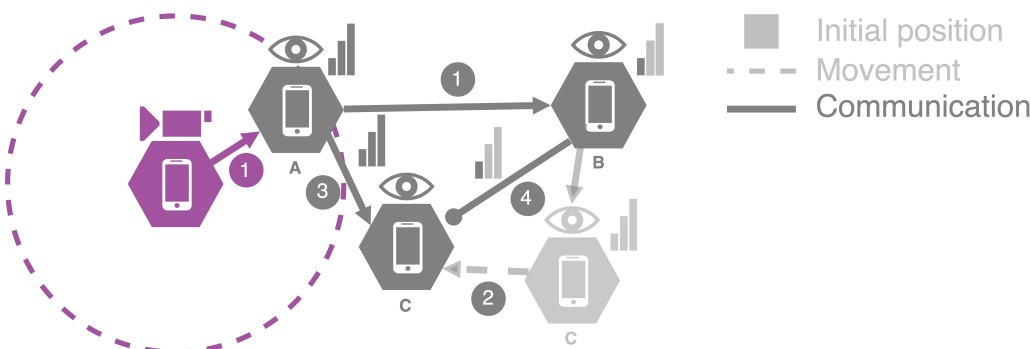

**Figure 5.** Routing strategy applied between intermediate nodes.

This strategy facilitates dynamic connections between devices within the MANET, leveraging knowledge and predictions to improve the overall communication performance. Technically, Algorithms 1 and 2 present the details of the routing strategy. More specifically, Algorithm 1 is executed when the sender begins to broadcast the video stream and

ends when the transmission is closed. To handle this communication process, a thread is launched when transmission is opened and stopped when the node finishes it. Thus, once the process begins, the device checks the surrounding connections and pings them to evaluate the connection metrics. At the same time, Algorithm 2 is launched when the receiver node becomes a discoverable node in the MANET. Likewise, this process may be interrupted by the node, stopping the reception and the response to incoming pings.

During the execution period, in Algorithm 2, receiver nodes respond to incoming ping requests only if they are not directly receiving the stream from the sender or if their current link to another node is unstable. Meanwhile, in Algorithm 1, senders manage this scenario with timeouts, allowing them to stop waiting for a response. If pinged nodes are not connected to any node or their current link is deemed unstable, they reply to the ping, thereby enabling senders to assess communication metrics and predict link stability with their pre-trained classification model. Subsequently, the sender communicates the prediction results to the receiver, which then evaluates whether the stability of the new link is superior to the existing one. If the assessment is positive, the node discontinues the current link, manages encryption as described in Figure 4, and starts receiving the stream from the new sender. If not, no changes to the link are made.

---

**Algorithm 1** Routing strategy in sender node to prioritize streaming to stable connections

---

**Require:** *executeBroadcast()* as the execution condition for the thread in charge of managing video streaming transmission, *I* as communication interface, *C* as current alive connection, $\Gamma$ as the trained model to predict link stability.

1:   *I.incomingPetitions()*              ▷ Incoming petitions are handled concurrently.
2:   **while** *executeBroadcast()* is true **do**
3:        *I.scan()*
4:        $N \Leftarrow I.scannedDevices() \setminus C$
5:        **while** $N \neq \varnothing$ **do**
6:            ping N.getDevice()
7:            initResponseMetrics()
8:            **if** I.receivePingResponse() is true **then**     ▷ Timeout to handle lack of response.
9:               $M \leftarrow calculateResponseMetrics()$
10:             $R \leftarrow \Gamma.predict(M)$
11:             I.transmitPrediction(R)
12:             **if** I.receiveConfirmation() is true **then** ▷ Timeout to handle lack of response.
13:               interchangePublicKeys()   ▷ Interchange public keys with the other node.
14:               shareSymmetricKey()
15:               retransmitStreaming()
16:               N.getDevice() into C
17:             **else**
18:               End iteration
19:             **end if**
20:            **else**
21:              End iteration
22:            **end if**
23:        **end while**
24: **end while**

---

This strategy outlines a method which addresses the variability of links in MANETs, aiming to sustain communication stability. Here, ping processes play a vital role in identifying contextual variables, thus enabling a distributed, self-governed approach to video streaming retransmission. At the heart of the proposal is the ML-based classification model. The following subsection will elaborate on the key features involved in link stability prediction.

---

**Algorithm 2** Routing strategy in receiver node to prioritize streaming from stable connections

---

**Require:** *executeReception*() as the execution condition for the thread in charge of managing incoming communications, *I* as communication interface, *C* as current alive connection, *N* as new incoming communication, $\alpha$ as sender device in the MANET, $\omega$ as stable communication.

1: **while** *executeReception*() is true **do**
2:     *I.scan*()                        ▷ Scanning process is handled concurrently.
3:     **if** *C.incomingPetition*() is true **then**
4:         **if** *I.otherEnd* is not $\alpha$ **then**
5:             **if** *I.predictedStability* is not $\omega$ **then**
6:                 *C.confirmPing*()     ▷ Waits for predicted stability from other end.
7:                 **if** *C.predictedStability*() > *I.predictedStability*() **then**
8:                     *C.confirmConnection*()
9:                     *I.closeCurrentConnection*()
10:                     interchangePublicKeys()     ▷ Interchange public keys with the other node.
11:                     receiveSymmetricKey()
12:                     *I.connect*(*I*)
13:                 **end if**
14:             **end if**
15:         **end if**
16:     **end if**
17: **end while**

---

### 3.4. ML Classification Model

The architecture incorporates a pre-trained ML classification model designed to forecast the stability of network links. To achieve this, the model utilizes a collection of connection attributes both for training and for making predictions. As outlined in Table 1, the features include RSSI, throughput, BER, number of connection attempts, latency, and distance. These parameters equip the model with essential and streamlined insights into the potential stability of the link. For the purpose of classification, two distinct methodologies can be adopted: binary and multi-class classification, each offering a different level of granularity in assessing link stability.

In the context of binary classification, two primary categories can be distinguished: stable links and unstable links. This distinction allows video streams to be adapted to two compression modes: highly compressed and lightly compressed. On the other hand, multi-class classification offers a more detailed level of granularity, enabling the categorization of links into a broader range of options. This approach contributes to a more precise identification and classification of link stability, facilitating tailored streaming adjustments.

The classification predictions offered by the model necessitate establishing a definitive hierarchy to assess the stability of the links. This enables prioritization of connections that demonstrate superior performance. Similarly, training the models demands datasets rich enough to deduce the stability of these links. Two methodologies can be pursued in this regard: one involves labeling connection sets for supervised learning and the other relies on using unlabeled communication datasets, steering the training towards unsupervised learning. The choice between these methods depends on the context's specific nature and the availability of the data. Moreover, given that the architecture is intended for deployment on mobile nodes, federated learning [31] emerges as a viable strategy for managing model updates and facilitating iterative enhancements.

Federated learning is a ML approach that involves deploying a pre-trained model across a set of nodes for individual predictions, which may include further training with newly introduced data [31]. This method allows for the consolidation of advancements from various models into a central model, striving for a comprehensive understanding of all individual models' insights. Additionally, the synchronization processes can be executed when devices have Internet access, facilitating autonomous operation in offline scenarios.

Therefore, federated learning introduces an additional layer that could potentially enhance the overall performance of the architecture.

Leveraging the potential of the proposed model, the architecture integrates ML as a pivotal element to enhance video streaming performance in MANETs and serves as a foundation for adaptive streaming. Proof of this concept has been developed to assess the technical feasibility and performance of the classification model, in addition to analyzing QoS and QoE metrics derived from the testing phase.

## 4. Results

To evaluate the proposal presented in this paper, a proof-of-concept application for smartphones was developed. This implementation facilitates the examination of streaming transmission performance, adhering closely to the outlined architecture. A variety of contexts were established to conduct comprehensive assessments, focusing on evaluating model performance as well as the QoS and QoE outcomes. This section presents the testbed scenario, describes how the model was trained, and provides an analysis of QoS and QoE.

### 4.1. Testbed Scenario

The testbed has been carried out in a laboratory context, utilizing a proof-of-concept implementation that enables the transmission of real-time video streaming in MANETs (https://github.com/ManuJazz/StreamingMANETs, accessed on 16 April 2024). This is achieved by using Bluetooth Low Energy (BLE) for context recognition and Bluetooth Classic for the transmission of frames between nodes. This technology was chosen due to its capability to broadcast information between smartphones without the need for prior pairing. Unlike Wifi Direct [32], no prior configuration is necessary, enabling opportunistic broadcasting of information. Also, encryption and decryption processes were not considered for the proof of concept.

The developed application was installed on three physical smartphones: one device had the task of capturing real-time video and two phones were responsible for receiving, screening, and routing the stream. These nodes were engaged in three different scenarios. The experiments were conducted in a laboratory context, where the physical positions of the devices were changed. This setup allows for the assessment of the proof-of-concept implementation's performance under multiple network conditions. Consequently, as Figure 6 illustrates, three distinct contexts ($C$) were considered: a first context in which receiver nodes are in contact with the sender ($C_a$); a second context in which one of the receiver devices moves to the edge of the communication range of the sender ($C_b$); and a third context in which a receiver device moves out of the sender's range, requiring the other receiver device to act as an intermediate node by routing the streaming ($C_c$). This enables the evaluation of the impact of the dynamic topology on the architecture and the resilience of the adaptive video streaming and routing strategy.

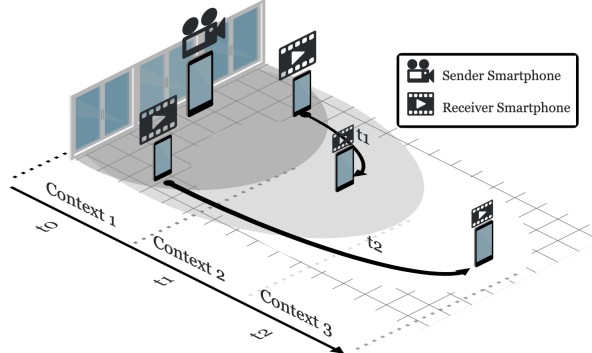

**Figure 6.** Testbed scenario with the distribution for each evaluation context.

A sequence of four iterative tests was performed for each scenario to minimize potential deviations. Before conducting these tests, the model integrated into the proof of concept

was trained with real-world data on interactions in MANETs. The following subsection will provide a detailed analysis of this process.

*4.2. Training the Model*

The model was trained using a dataset generated by a testbed tool designed to assess QoS metrics in MANETs [33]. This tool, installed on smartphones, is dedicated to providing comprehensive reports on the performance of physical devices within a Bluetooth MANET environment. For this study, data were generated in a testbed MANET configuration involving five dynamic smartphones. The reports generated by the tool registered interactions, contributing to the formation of an accurate dataset and including the attributes detailed in Table 1. The dataset includes self-tagged interactions to indicate the success of communications between devices: true interactions were those in which the flows of ping messages were successfully received back by the sender, and false interactions occurred when communication was interrupted by either end. These values can be applied to identify the communication variables which encourage stable connections (tagged as true) and those values which may become indicators of unstable connections (tagged as false). This binary approach was chosen over multi-class classification primarily for its accuracy in terms of labeling data. Consequently, this feature facilitates the dataset's applicability for training the model.

As an initial phase, we curated the dataset to remove noise, such as entirely empty rows and duplicates. The resulting dataset consists of 5257 rows and 7 attributes (Table 1), including one additional column indicating whether the communication was successful. To split the dataset, we applied a distribution of 70–30% for training and validation, respectively. This division is a common practice in machine learning to ensure a balance between training and validation [34], resulting in 3680 rows for training and 1577 rows for validation.

The considered data collection enables the application of binary classification algorithms. Three different algorithms have been considered: Two-Class Decision Forest, Logistic Regression, and Two-Class Neuronal Network. These alternatives have become some of the most relevant techniques for ML-based classification [35]. As depicted in Figure 7, the sensitive analysis reflects that Two-Class Decision Forest provides the best overall results, outperforming alternatives. These results include accuracy (0.959), precision (0.935), recall (0.872), and F1 Score (0.903). The results indicate that the model successfully predicted most of the evaluation instances. The high accuracy score demonstrates effective classification in the majority of cases. Precision highlights that stable links were correctly identified in 93.5% of instances. The recall rate, closely mirroring precision, shows the model's ability to accurately flag most unstable connections. Lastly, the F1 Score, representing the balance between precision and recall, attains a high value, reinforcing confidence in the model's performance.

Regarding the configuration of the Decision Forest algorithm, the model was trained using bagging as the resampling method, with a total of eight decision trees. Furthermore, each tree was restricted to a maximum depth of 32. This configuration was designed to mitigate overfitting during the training process. As a result, the training data facilitated the model's training, adhering to a set of predefined configurations.

The input values proposed for the model are part of a strategy designed to achieve high accuracy in optimization, with a low tolerance value ($8 \times 10^3$) and elastic net parameters for L1 and L2 weights. After training, it was observed that attributes such as distance, disconnections, BER, and latency are assigned negative weights, indicating their inverse impact on stability classification. Conversely, the RSSI has been identified as having a direct positive impact on the outcomes, while parameters like throughput are not deemed significantly relevant.

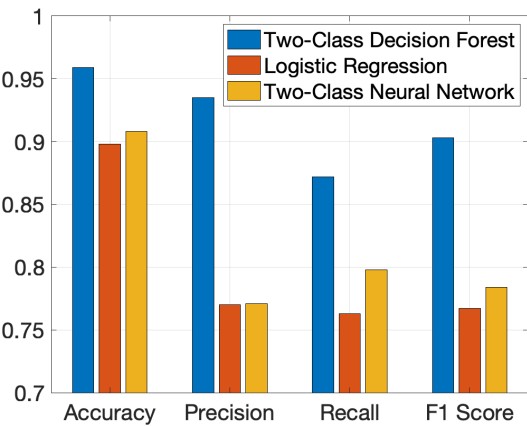

**Figure 7.** Performance metrics comparison for the classification algorithms.

As a result, the global evaluation score of the model is positive and encouraging. Predicting the stability of links in MANETs is a challenging task, since the network may behave randomly. However, the obtained results highlight a significant performance.

*4.3. Analysis of QoS and QoE*

During the execution of the application in the different contexts, QoS and QoE parameters were used to evaluate the impact of adaptive video streaming. Aiming to compare the obtained results objectively, latency has also been measured, applying fixed-compression transmission. Furthermore, latency values were also considered, due to its significant relevance in the communication. As a result, Figure 8 represents the resulting values of latency for the adaptive video streaming (Adapted Streaming, AS) and for fixed-compression streaming (Fixed Streaming, FS), including individual executions of the three contemplated scenarios.

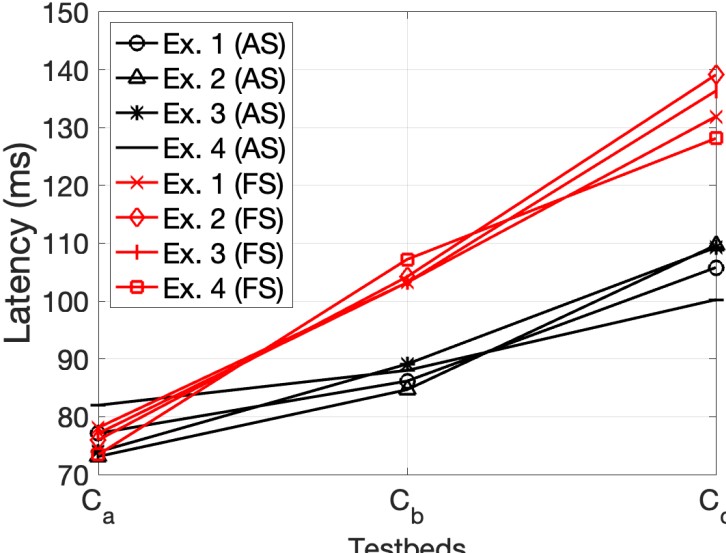

**Figure 8.** Obtained latency values for the executions of Adaptive Streaming (AS) technique and Fixed Streaming (FS) technique, at the different contexts ($C_1$, $C_2$ and $C_3$).

Figure 8 displays lower latency values for adaptive streaming. It is possible to observe an increasing tendency as the distance between sender and nodes increases ($C_b$ and $C_c$). However, dynamic frame compression improves fixed compression, easing transmission and improving latency. As a result, it is possible to observe a global stable tendency which is viable for video streaming and resilient to node movements.

On the other hand, QoE can be assessed under variables such as the resolution of the received images. In this context, the proof of concept compresses the frames by around 40% if the link is identified as unstable. Thus, Figure 9 displays two frames extracted from the live video, with one corresponding to a low compression rate and another with a high compression rate.

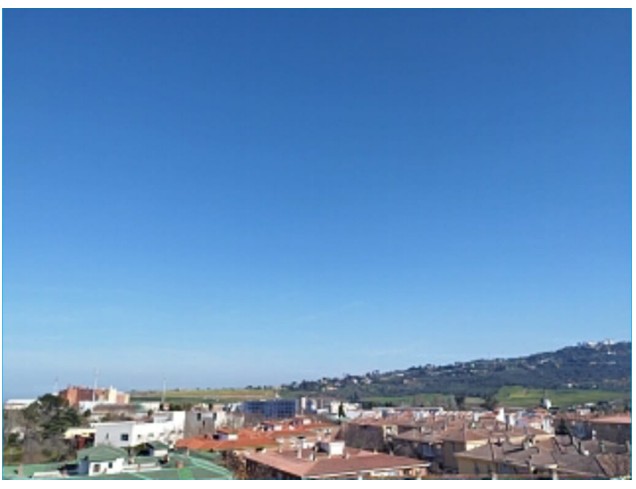

(**a**)

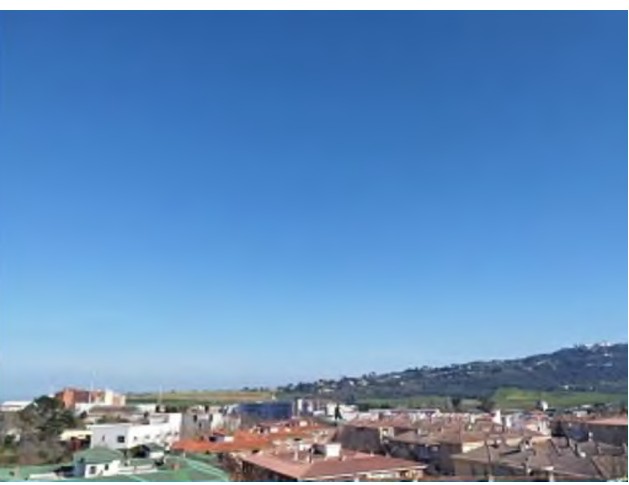

(**b**)

**Figure 9.** Compression frame comparison for stable and unstable link. (**a**) Light compressed frame for stable link. (**b**) High compressed frame for unstable link.

In this visual comparison, differences in the quality of the images are noticeable. However, the loss is not pronounced in terms of shape identification or resolution but rather in terms of colors and sharpness. In Figure 9a,b, variations in color tones are observable, particularly in the blue color, clouds, and mountains in the background. Nonetheless, buildings are discernible in both frames in a very similar manner. As a result, variations in compression are not immediately obvious to the viewer. Considering these results, the next section discusses the main findings of the applied approach.

*4.4. Discussion*

The implemented proof-of-concept establishes a robust foundation for applying the described architecture in real-world scenarios. The results obtained yield promising outcomes for the ML model, as well as for QoS and QoE, thereby forming a comprehensive approach.

Firstly, the trained classification model achieves high evaluation scores, demonstrating significant effectiveness in accurately classifying links based on their stability. Given that

the training process was conducted using actual results from a physical MANET, the results obtained are likely reliable for application within the context of the proposed architecture. In this manner, the reduction in latency (Figure 8) can be aligned with the accuracy achieved in model validation. Similarly, as depicted in Figure 9, the binary classification does not significantly impact QoE. However, some limitations in the obtained results should be acknowledged.

The ML model was trained with data collected from physical smartphones, which often execute multiple tasks simultaneously, in addition to ad hoc communications. This concurrent usage of computational resources can influence communication metrics, potentially leading to additional delays in the calculation of latency or BER. Since such factors are inherent in real MANET environments, the impact of these variations can be mitigated by including data from a more extensive array of devices in the dataset generation, thereby minimizing any introduced deviations.

Secondly, the latency values demonstrate the positive impact of adaptive streaming on the QoS in MANETs. When compared to the results from fixed streaming, this improvement can be attributed to the adaptation in the size of the frames. However, it is crucial to note that the experimental setup primarily explores the effectiveness of streaming. Moreover, there are other significant metrics, especially those concerning network scalability. To thoroughly assess this, future experiments should involve a larger number of smartphones and contexts.

Regarding the user's perspective, changes in compression have no discernible effect on the appearance. Therefore, even if there is a shift in link classification while video streaming is in progress, the variations in quality are not drastic.

Reflecting on the discussion of the results obtained from the experiment, the findings encourage further exploration and enhancement of the proposal in future research endeavors.

## 5. Conclusions and Future Work

In recent years, live video streaming in MANETs has emerged as a frequent area of research, propelled by the distinct challenges inherent to these networks. The dynamic nature of MANETs, characterized by unstable and unpredictable topology and potential disconnections between nodes, poses considerable obstacles to maintaining QoS during live broadcasts. This paper presents a software-based architecture designed to support adaptive streaming in MANETs through the application of ML techniques. The aim of this strategy is to mitigate the adverse impact of MANETs' challenges, thereby ensuring a more stable and reliable live video streaming experience in MANET environments.

This paper outlines a proposal that employs ML-based classification techniques to assess the potential stability of links between nodes. Based on these predictions, video streaming frames are compressed accordingly, aiming to improve QoS by optimizing the size of video frames. Considering this design, the proposed approach has been implemented and evaluated in a physical MANET using a decision tree classification model and Bluetooth as the communication interface. As a result, the model achieves an accuracy of 95.9% and an average latency ranging from 75 to 106 ms. Additionally, adaptive compression does not critically affect quality, enabling smooth transitions if the link becomes unstable.

Considering the proposal and the results obtained from the evaluation, the architecture emerges as a promising approach to implementing distributed solutions that necessitate video streaming in environments that lack Internet access. This scenario is particularly pertinent in Dew Computing networks, in which multiple nodes collaborate to perform tasks offline. Within this context, the architecture is designed to facilitate video transmission, thereby enabling Dew nodes to perform ML-based services, such as visual pattern recognition in incoming videos, transcription of detected text, or screening streaming. Since the architecture aims to provide minimized latency and reduced interruptions, these functionalities may significantly enhance its performance. This capability makes adaptive

streaming a valuable resource for improving the efficiency and reliability of operations carried out by Dew nodes.

Future research will focus on improving the implementation of the strategy, specifically addressing the limitations identified in the proof-of-concept implementation, especially scalability and security. On the one hand, scalability may be enhanced by refining the routing policy, taking into account the workload of sender nodes as a constraint. This way, routing based on the availability of resources can be implemented, aiming to offload senders by utilizing receiver nodes as intermediate nodes, even in scenarios in which nodes are in direct contact with the sender. Similarly, the binary classification of links could be enhanced by adopting multi-class classification. This approach is expected to achieve a higher granularity in predictions, aiming to more effectively adapt compression to the state of communication. On the other hand, the integration of encryption processes and the study of its impact on QoS and QoE will become a relevant part of future studies. As a result, the present work establishes a solid initial approach toward advancements in video streaming in offline MANETs.

**Author Contributions:** Conceptualization, M.J.-A., V.N.G.J.S. and J.G.-J.; software, M.J.-A., V.N.G.J.S. and J.G.-J.; validation, M.J.-A., V.N.G.J.S. and J.G.-J.; funding acquisition, V.N.G.J.S. and J.G.-J. All authors have read and agreed to the published version of the manuscript.

**Funding:** This work has been partially funded by MCIN/AEI/10.13039/ 501100011033 and by the European Union "Next GenerationEU /PRTR", by the Ministry of Science, Innovation and Universities (projects TED2021-130913B-I00, PDC2022-133465-I00), European Union's Digital Europe Programme (DIGITAL) under grant agreement Nº101083667-TECH4EFFICIENCYEDIH. It is co-funded by the Ministry of Industry, Trade and Tourism of the Government of Spain, within the framework of the Recovery and Resilience Mechanism. However, the views and opinions expressed are solely those of the author(s) and do not necessarily reflect those of the European Union or the Government of Spain. The authors also acknowledge the Science and Digital Agenda of the Regional Government of Extremadura (GR21133) and the European Regional Development Fund. V.N.G.J.S. acknowledges that this work is funded by FCT/MCTES through national funds and, when applicable, co-funded EU funds under the project UIDB/50008/2020.

**Data Availability Statement:** The original data presented in the study are openly available in Github at https://github.com/ManuJazz/StreamingMANETs, accessed on 16 April 2024.

**Conflicts of Interest:** The funders had no role in the design of the study; in the collection, analyses, or interpretation of data; in the writing of the manuscript; or in the decision to publish the results.

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
