# Peer review of "ML-Enhanced Live Video Streaming in Offline Mobile Ad Hoc Networks: An Applied Approach"

_electronics, doi:10.3390/electronics13081569_

Round 1
Reviewer 1 Report
Comments and Suggestions for Authors
Summary: The paper aims to enhance live video streaming in Offline Mobile Ad-Hoc Networks (MANETs) using machine learning techniques. The main contributions include proposing a novel architecture for dynamic and individual adaptability of streaming, integrating software-based components for communication management, link prediction, and frame forwarding. The strengths lie in the detailed analysis of the architecture, the training of the ML model, and the evaluation of QoS and QoE metrics.
General Concept Comments: The paper effectively addresses the challenges of live video streaming in MANETs and proposes a comprehensive solution. However, there could be more emphasis on the limitations of the proposed technique and a comparison with existing state-of-the-art methods. Additionally, the paper could benefit from a more detailed discussion on the scalability and real-world applicability of the proposed architecture.
Suggested improvements::
- In Section 3.2, it would be beneficial to provide more insights into the scalability of the proposed architecture, especially in scenarios with a large number of nodes.
- In Section 4.2: The training process of the ML model could be described in more details to provide clarity on the dataset used and the specific algorithms employed
- The evaluation metrics in Section 4 could be more detailed, including a discussion on the limitations of the model and potential areas for improvement.
- In Section 5: The limitations and future direction should be presented in more structured method.
Comments on the Quality of English LanguageEnglish level pretty well
Author Response
Please find in the attached PDF file a point-by-point response to your comments and suggestions. We greatly appreciate your remarks, as they help improve our work. Thank you very much.

Reviewer 2 Report
Comments and Suggestions for Authors
Dear authors, congratulations on the research, it was very pleasing to read the manuscript. Please consider to improve your research with the following recommendations:
1) Page 2: order the ordinal numbers of references, instead of "[11][9]", use "[9][11]":
2) Page 4: it would be great to extend current figures 1 and 2 (and 4) with ordinal numbers describing the order of actions that happened in the described scenarios. For example, identifying nodes could be marked with the number "1" in the first action.
3) Page 5: please provide more details about the compression algorithm you mentioned in step 3.
4) Page 6: please provide more details about the communication security among nodes.
5) Page 7: you have an indefinite "while" loop, but it's not clear where this loop ends, with something like a "break" statement or similar: ""
Please describe in more detail if this is active all the time, or ends at a specific moment. The remark counts for Algorithm 2 as well.
6) Please explain all used abbreviations in the paper, like RSSI, BER, etc. on the first mention, like on page 11:
7) Please provide more details about the dataset used in the ML model, size, division to training and validation data set parts, etc.
8) Please provide the source code as supplementary material to the manuscript.
9) Please reflect in the conclusion section on other scenarios where this methodology could be reused.
Author Response

(The authors gave the same response as above.)

Round 2
Reviewer 1 Report
Comments and Suggestions for Authors
The authors addressed all my comments.
I have no further comments.
Comments on the Quality of English LanguageEnglish level is quite good, no